# Soil fungal communities differ between shaded and sun-intensive coffee plantations in El Salvador

**Maya V. Rao[1,2], Robert A. Rice[3], Robert C. Fleischer[2], Carly R. Muletz-Wolz[2]***

**1** Department of Biology, University of Maryland, College Park, MD, United States of America, **2** Center for Conservation Genomics, Smithsonian National Zoological Park & Conservation Biology Institute, Washington, DC, United States of America, **3** Migratory Bird Center, Smithsonian National Zoological Park & Conservation Biology Institute, Washington, DC, United States of America

\* muletzc@si.edu

**Data Availability Statement:** Demultiplexed pyrosequencing run sequence data and associated metadata has been deposited in the National Center for Biotechnology Information Sequence Read

## Abstract

*Coffea arabica* is a highly traded commodity worldwide, and its plantations are habitat to a wide range of organisms. Coffee farmers are shifting away from traditional shade coffee farms in favor of sun-intensive, higher yield farms, which can impact local biodiversity. Using plant-associated microorganisms in biofertilizers, particularly fungi collected from local forests, to increase crop yields has gained traction among coffee producers. However, the taxonomic and spatial distribution of many fungi in coffee soil, nearby forests and biofertilizers is unknown. We collected soil samples from a sun coffee system, shade coffee system, and nearby forest from Izalco, Sonsonate, El Salvador. At each coffee system, we collected soil from the surface (upper) and 10 cm below the surface (lower), and from the coffee plant drip line (drip line) and the walkway between two plants (walkway). Forest soils were collected from the surface only. We used ITS metabarcoding to characterize fungal communities in soil and in the biofertilizer (applied in both coffee systems), and assigned fungal taxa to functional guilds using FUNGuild. In the sun and shade coffee systems, we found that drip line soil had higher richness in pathotrophs, symbiotrophs, and saprotrophs than walkway soil, suggesting that fungi select for microhabitats closer to coffee plants. Upper and lower soil depths did not differ in fungal richness or composition, which may reflect the shallow root system of *Coffea arabica*. Soil from shade, sun, and forest had similar numbers of fungal taxa, but differed dramatically in community composition, indicating that local habitat differences drive fungal species sorting among systems. Yet, some fungal taxa were shared among systems, including seven fungal taxa present in the biofertilizer. Understanding the distribution of coffee soil mycobiomes can be used to inform sustainable, ecologically friendly farming practices and identify candidate plant-growth promoting fungi for future studies.

Archive (www.ncbi.nlm.nih.gov/sra) under
BioProject ID: PRJNA558046. Final files for
analysis (feature table, taxonomy table,
phylogenetic tree and metadata file) and R code are
available from figshare: https://doi.org/10.6084/
m9.figshare.9209390.

**Funding:** The author(s) received no specific
funding for this work.

**Competing interests:** The authors have declared
that no competing interests exist.

## Introduction

Cultivation of *Coffea arabica* has gained notoriety for its important role in impacting local and global biodiversity [1]. Coffee plantations with shaded tree canopies provide refuge for a wide variety of plant [2, 3] and animal species, including herbivorous insect species [4–6], temperate-tropical migratory birds [7–10], amphibians [11] and mammals [12–14]. Within just one gram of soil, there exists a staggering number of up to 38,000 different bacterial taxa [15]. Coffee soil also contains a high diversity of fungi, particularly arbuscular mycorrhizal (AM) fungi, which are highly valued because of their beneficial effects on plant growth and their ability to reduce water and nutrient losses [16]. Quantifying overall fungal diversity, including AM fungi as well as other symbiotic fungal taxa, in coffee systems provides insight into plant-microbial relationships, which can inform sustainable agriculture practices [17].

Coffee is one of the most highly traded and valued commodities that originates from the developing world [18], with global consumption increasing nearly twenty-fold since 1952 [15]. To meet the demands, coffee cultivation practices have shifted from a traditional shade system to a sun-intensive system starting in the 1970s [2]. Traditional shade coffee farms ('agroforestry' systems) are characterized by their inclusion of tall tree species that provide forest-like shade [19]. In contrast, modern sun systems remove nearby trees and rely heavily on chemical inputs, pruning, and high coffee plant density to increase yields [2, 20].

The implementation of modern systems increases coffee yield in comparison to traditional systems [2], but results in a loss of the valuable canopy cover found in traditional coffee farms. Differences in canopy coverage between sun and shade systems affects local temperatures and humidity, which in turn affects environmental suitability for local flora and fauna. For instance, the monthly average maximum temperatures of a full sun coffee system in the Brazilian Atlantic Rainforest was 5.4˚C higher than another coffee agroforestry system practicing a traditional shade approach [21]. The lack of canopy foliage also causes modern plantations to become more prone to water and soil runoff, which in turn threatens the sustainability and biodiversity of the system [22]. Perfecto et al. [2] found that traditional coffee plantations have a high structural complexity that offers living and nesting sites for a large variety of organisms, ranging from plants to amphibians to parasites. In contrast, modern sun coffee plantations lack the protection conferred by canopy foliage and the input of leaf litter, which reduces soil moisture and removes an additional layer of habitats for organisms across multiple kingdoms [23].

Microbes play integral roles in agriculture from pathogenic to beneficial partners of agricultural plants. Microbial communities, including bacteria and fungi, affect the structure and ecological roles of plants by altering various functional traits of plants [24]. Beneficial bacteria and fungi occur widely. Mycorrhizal fungi are known for their symbiotic relationships with over 90% of all plant species [16, 25], and are particularly known for facilitating the efficient uptake of nutrients such as phosphorus [26]. On the other hand, fungi also have the potential of causing disease. Coffee leaf rust, *Hemileia vastatrix*, and coffee berry disease, *Fusarium xylarioides*, are just two of the major fungal pathogens that are causing widespread damage to tropical plants like *Coffea arabica* [27]. Using local plant-associated microorganisms to increase crop yields, and to reduce economic and environmental impacts of pesticides and monoculture-focused farming has gained traction among coffee producers in recent years. Fungal agents collected from local forests and then used in biofertilizers play a central role in these efforts. Coffee management practices can impact fungal diversity and crop output [16, 17, 28], which have implications for best management practices in sustainable agriculture.

We characterized fungal diversity in soils from a sun-intensive coffee plantation, traditional shade coffee plantation and nearby forest as well as from a biofertilizer applied in both coffee

plantations. We collected samples from organic coffee plantations found in Sonsonate, El Salvador. El Salvador is the world's eighth largest coffee producer, and coffee, along with sugar and cotton, contribute to 75% of the country's export earnings [29]. As farmers in El Salvador turn toward sun-intensive coffee systems to increase output and efficiency, there is a need for analysis of the environmental and ecological implications these agricultural decisions entail. Microbes can be used as biocontrol agents as they have the capacity to mitigate and decrease pathogenic colonization of harmful microbes [30], likely by decreasing the competitive ability of pathogens [31]. We had three main objectives. First, quantify the impacts of microhabitat (upper vs. lower soil depth, walkway vs. drip line of plant) and habitat (sun vs. shade coffee farms) on fungal diversity. Second, identify function of the fungi and determine how functional guilds vary across these microhabitats and habitats. Third, quantify differences in fungal diversity and functional guilds across sun coffee soil, shade coffee soil, and soil from a nearby forest with no coffee plantation. Our study offers insight into the taxonomic and ecological differences in fungal species of the coffee plant soil between sun intensive and traditional shade coffee cultivation practices in organic farming. As Juan José Paniagua, a pioneer in organic farming in Costa Rica, said "El suelo es un mundo do los seres más pequeños, ellos lo cuidan para nuestra esperanza" or "the soil is a world of the smallest beings, they care for it for our hope."

## Materials and methods

### Field sampling

We collected samples of shade and sun-intensive coffee (*Coffea arabica*) soil from the ACOPRA Las Lajas Coffee Cooperative in Izalco, Sonsonate, El Salvador and from a nearby forest. The land is privately owned by the ACOPRA Las Lajas Coffee Cooperative and we received permission from them to collect the soil samples. We received Both the sun and shade systems are certified organic farms and are treated with the same biofertilizer and compost. They receive applications of a fungus-based liquid biofertilizer (Biofertilizante Engruese, NCBA CLUSA) that is used as a foliar spray and injected into the soil of the coffee plant at least two to three times per year, as well as a compost generated with the Japanese 'Bocashi' process that consists of coffee pulp, animal manure, and vegetation. The base of the fertilizer is an unknown fungus (referred to as 'mountain fungus', S1 Fig) that is collected in the forest from the leaf litter layer and mixed with molasses, whey, rock phosphate and other ingredients to create the final product. The sun coffee system site was located at a latitude of 13º 48' 27.3" N, a longitude of 89º 35' 57.3" W, and at an elevation of 1227 meters. The shade coffee system site was located at a latitude of 13º 49' 7" N, a longitude of 89º 34' 54" W, and at an elevation of 1050 meters. The forest site was located at a latitude of 13º 49' 41.5" N, a longitude of 89º 34' 10.3" W, and an elevation of 907 meters. All sites were within 1 kilometer of one another, which mitigates some of the issues associated with one replicate site per system. We randomly selected seven coffee bushes in each system (sun and shade) to sample. We took one soil sample within the drip line of each coffee bush (drip line samples), and another soil sample from outside the drip line between the coffee bush and the adjoining row of coffee bushes (walkway samples). Each drip line and walkway spot had a surface sample of the mineral soil (upper) and a sample taken at 10 cm depth (lower). We collected 14 surface soil samples from the nearby forest. In total, we had one sample of the biofertilizer and 70 soil samples (Fig 1; 28 samples from seven sun coffee bushes, 28 samples from seven shade coffee bushes, 14 samples from a nearby forest).

For each soil surface sample, we collected approximately three grams of soil with a spoon that was wiped cleaned and alcohol sterilized between each sample collection. To collect the 10

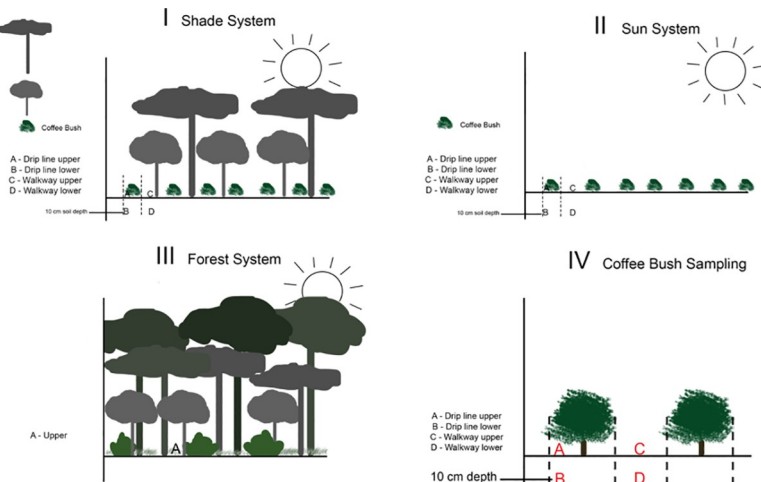

**Fig 1. System and bush sampling schematic.** I & II. We randomly sampled soil from seven bushes from each of the coffee plantation systems (shade and sun) with four sampling sites per bush. III. We randomly collected 14 soil samples from a nearby forest. IV. Zoomed in view of II: we collected an upper drip line sample, 10 cm depth drip line sample, upper walkway sample, and 10 cm depth walkway sample for each sun and shade coffee bush.

cm depth samples, we used a soil probe using sterile plastic tubes with one tube per sample. All soil samples were placed into a sterile Whirl-Pak bag labelled with a unique ID. Soil samples were stored in a conventional freezer until transferred on dry ice to the Center for Conservation Genomics (National Zoological Park, Washington, DC). Samples were then stored in a -80º C freezer until we performed DNA extractions. We had permission from the US Department of Agriculture Animal and Plant Health Inspection Service (permit # P330-17-00090) to import the soil samples.

## Molecular methods

We extracted DNA from the biofertilizer and soil samples using the Qiagen DNeasy PowerSoil Kit by following the manufacturer's protocol. A negative extraction control was included with each set of extractions. We used the universal gene primer ITS86F and ITS4 [32] to amplify the large subunit ITS2 region and 5.8S gene of fungal taxa. We performed duplicate PCR reactions for each soil sample, and included negative extraction controls and negative PCR controls. The 20 μl individual PCR assays consisted of 10 μl of 2x Phusion Hot Start II HF Master Mix, 400 nM reverse primer, 400 nM forward primer, and 2 μl DNA template. PCR conditions were 98˚C for 30 s, followed by 25 cycles of 98˚C for 10 seconds, 58˚C for 20 seconds, 72˚C for 30 seconds and a final extension (72˚C for 5 m). We performed index PCR, adding nextera-style i5 and i7 adaptors to the PCR amplicons to uniquely identify each sample. We performed post-PCR clean-ups between each PCR reaction using [33] Speed-beads (in a PEG/NaCl buffer). We quantified the molarity of each final library-prepped sample, and then pooled samples in equimolar amounts [34]. We used Illumina MiSeq High-Throughput sequencing (2 x 300 bp kit) to characterize the soil fungal communities of each sample.

## Sequence analysis and diversity calculations

We used QIIME 2.0 [35] to process the reads and R version 3.5.2 to calculate diversity indexes. In QIIME, we assigned reads to samples based on unique i5s and i7s barcodes and filtered out low quality reads using dada2 [36] with the following criteria:—p-trunc-len-f 260—p-trunc-len-r 200—p-trim-left-f 21—p-trim-left-r 20. Sequences were then categorized into amplicon

sequence variants (ASVs) via the dada2 pipeline and taxonomy was assigned by aligning ASVs with the UNITE database (version 7) [37]. A phylogenetic tree was built using the fasttree algorithm [38]. Files created in QIIME were exported and then imported into R, where all subsequent analyses were conducted. We examined extraction and PCR negative controls for contamination, and found that the ASVs detected in the negative controls were unique to the controls; those ASVs and negative extraction controls were removed from subsequent analyses. We removed all ASVs that were not assigned to fungi at the kingdom level.

We first conducted a general descriptive analysis of the fungal ASVs identified from the three systems and the biofertilizer by using the package 'phyloseq' to enumerate the relative abundances of dominant fungal classes. Then, we examined mycobiome structure in two subsets of the data: a sun vs. shade dataset and a sun vs. shade vs. forest dataset (Fig 1). In the sun vs. shade dataset, the sun and shade samples were taken from multiple locations (walkway and drip line) and multiple depths (upper and lower). In the sun vs. shade vs. forest dataset, the forest samples were taken only from the upper surface; in this dataset, sun and shade samples were subset to contain only the upper surface samples, including upper walkway and drip line samples.

We quantified alpha diversity using fungal ASV richness and Faith's Phylogenetic Diversity (Faith's PD) [39]. Faith's PD is the phylogenetic analogue of taxon richness and is expressed as the number of tree units which are found in a sample. We quantified fungal community composition (beta diversity) using Jaccard and Bray-Curtis distances. We used FUNGuild [40] to assign fungal ASVs to specific functions, which may include aiding in nutrient uptake and protecting against pathogens [41]. We uploaded an 'OTU file', which included taxonomic assignments for each ASV to the online classifier for FUNguild. ASVs were assigned to trophic modes (pathotrophs, symbiotrophs, and saprotrophs) if there was sufficient taxonomic information for the ASV. We included only "probable" and "highly probable" matches (discarding "possible" matches), and filtered for ASVs that exclusively were assigned as symbiotrophs, pathotrophs, and saprotrophs [42].

## Statistical analyses

All statistical analyses were conducted using R 3.5.2. Alpha diversity and beta diversity were used to examine the mycobiome structure of the coffee and forest systems. Alpha diversity consisted of overall species richness estimates and overall Faith's PD metric as well as ASV richness estimates for FUNGuild assigned pathotrophs, symbiotrophs, and saprotrophs. We analyzed the sun vs. shade dataset to determine effects of system, sampling depth, and sampling location on alpha and beta diversity. For alpha diversity, we used analysis of variance (ANOVA) to determine the effects of system, sampling depth, sampling location and their interactions on the total number of observed ASVs, Faith's PD, and pathotroph, symbiotroph, and saprotroph richness (in separate models). We conducted post hoc analyses for significant terms using the Tukey's 'Honest Significant Difference' (HSD) method. For beta diversity, we used a PERMANOVA to determine the effects of system, sampling depth, sampling location and their interactions on Jaccard and Bray-Curtis distance metrics. Prior to conducting Bray-Curtis analyses, we performed proportion normalization on the raw sequence counts to correct for biases associated with unequal sequencing depth on this abundance-weighted metric [43, 44]. Variation was minimal (3.7x difference in sequencing depth), so all other alpha and beta diversity metrics should be minimally impacted by sequence coverage [43, 44]. To visualize changes in ASVs and community composition (PCoA) between systems and locations, we used the packages 'phyloseq' and 'ggplot2' [45, 46]. We then analyzed the sun vs. shade vs. forest dataset to determine the effect of system on alpha and beta diversity as above with system

as the sole explanatory variable. We used the program Venny 2.1 [47] to make Venn diagrams of shared and unique ASVs among systems. Indicator species analysis was conducted to find the most characteristic ASVs from both the sun and shade. To do so we used the package 'indicspecies' with the multipatt function to identify ASVs that were almost exclusively found in sun and shade coffee soil samples, and generally absent from the forest soil samples [48]. We identified ASVs that had scores > 0.7 for both A (specificity) and B (fidelity) indicator values [48]. We report the ASVs that could be assigned to taxonomy below the phylum level from the QIIME2 taxonomy assignments (as reported above).

For fungal taxa with limited taxonomic information (e.g., only order level classification), we blasted individual sequences in NCBI using BLASTN 2.9.0+. to present greater taxonomic information. When all top matches (e-value < e-50) matched a fungus with high query coverage (> 99%) we recorded that fungal species as the taxonomic assignment.

## Results

We generated 3,061,400 high quality sequences from 70 samples representing 6,697 fungal ASVs and 9 fungal phyla. Average sequencing depth per biological sample was 43,734 sequences (min = 19,747, max = 73,593). The taxonomic composition of ASVs consisted predominantly of fungi in two phyla, Ascomycota and Basidiomycota. Ascomycota was represented by 4,696 ASVs and had an average relative abundance of 88.7% (SD ± 9.3) per sample. Basidiomycota was represented by 1,135 ASVs and had an average relative abundance of 8.7% (SD ± 8.2) per sample. The other phyla, which in total comprised less than 3% of average relative abundance per sample, included: Chytridiomycota, Entomophthoromycota, Entorrhizomycota, Glomeromycota, Mortierellomycota, Mucoromycota and Rozellomycota. Within Ascomycota and Basidiomycota, there were 8 dominant classes that were represented by > 1% of the total sequences (Fig 2). Sordariomycetes and Dothideomycetes accounted for a large portion of the classes found. Sordariomycetes had an average relative abundance of 45.1% (SD ± 15.6) per sample while Dothideomycetes had an average relative abundance of 19.5% (SD ± 11.7) per sample.

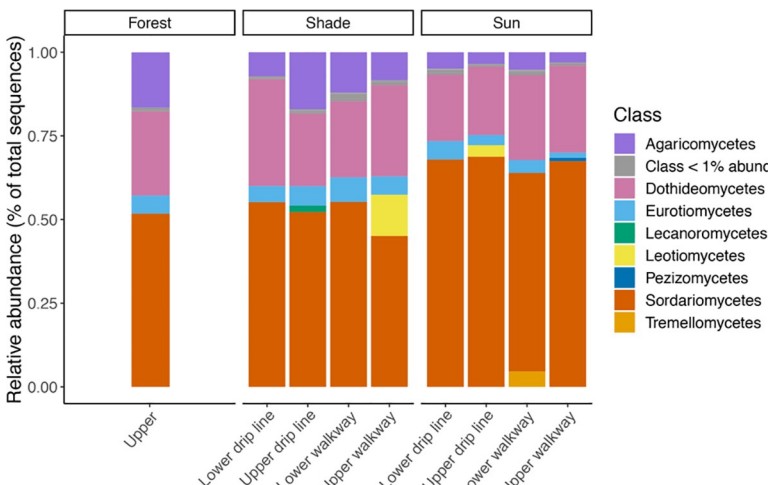

**Fig 2. Class abundance of soil fungi across systems, sampling depths, and sampling locations.** Eight dominant classes were found within the Ascomycota and Basidiomycota phyla which accounted for 97.2% relative abundance of the total sequences generated.

We identified 13 fungal ASVs in the biofertilizer from two fungal phyla (Table 1). For two fungal ASVs we were unable to resolve their taxonomic assignment beyond class level. Of the 13 fungal ASVs present in the biofertilizer, seven fungal ASVs were also found in coffee soil samples (Table 2) and only three were commonly detected in soil samples (Fig 3). We observed one dominant fungal taxa in the biofertilizer, a *Cylindrocladiella* sp., only occurring in the shade coffee system (Fig 3). The other two dominant fungal taxa present in the biofertilizer, a *Cladosporium* sp. and a Dothideomycetes sp., showed little sorting among systems and were present in most coffee soil samples (Fig 3).

## Sun vs. shade dataset

We quantified the effect of system (sun and shade), sampling depth (upper and lower), sampling location (drip line and walkway) and their interactions on alpha diversity and beta diversity. For alpha diversity, the only significant effect on fungal ASV richness and phylogenetic diversity was sampling location (ASV richness ANOVA, $F(1,48) = 4.8$, $p = 0.032$; Faith's PD ANOVA, $F(1,48) = 4.2$, $p = 0.046$). Specifically, we found higher fungal ASV richness (Fig 4A) and Faith's phylogenetic diversity (S1 Fig) in soil samples taken from the coffee bush drip line compared to those taken from the walkway. Sun and shade systems had similar fungal ASV richness and Faith's phylogenetic diversity as did upper and lower sampling depths (ANOVAs, $p > 0.05$). Average soil fungal ASV richness was 306 ASVs (Fig 4A, SD ± 88) and average Faith's phylogenetic diversity was 66 tree units (S2 Fig, SD ± 15). For beta diversity, the only significant effect on community composition was system with sun and shade coffee systems having distinct fungal communities (Fig 5: Jaccard: PERMANOVA, Pseudo F = 4.6, df = 1, $R^2 = 7.8\%$, $p = 0.001$; Bray-Curtis PERMANOVA, Pseudo F = 5.4, df = 1, $R^2 = 9.1\%$, $p = 0.001$). Sampling depth and location did not affect community composition (PERMANOVA, $p > 0.05$).

We tested the effects of system, sampling depth, and sampling location on the richness of pathotrophs, symbiotrophs, and saprotrophs (Fig 4B–4D). For pathotrophs and symbiotrophs, the only significant effect on their richness was sampling location (drip line and walkway). Pathotrophs and symbiotrophs richness was higher in the drip line compared to the walkway (pathotrophs ANOVA, $F(1,48) = 3.9$, $p = 0.055$; symbiotrophs ANOVA, $F(1,48) = 8.6$, $p = 0.005$). We found no effect of system or sampling depth on the richness of pathotrophs or symbiotrophs (ANOVAs, $p > 0.05$). For saprotrophs, we found significantly higher saprotroph

**Table 1. Fungal ASVs present in biofertilizer.** Fungal ASVs that were detected in coffee soils are in bold. Five ASVs could be categorized into a trophic mode using FUNguild and our criterion for assignment: three were 'probable' pathotrophs (ASV ID underlined) and two were 'probable' saprotrophs (ASV ID italicized).

| ASV ID | Phylum | Class | Family | Genus | Species |
|---|---|---|---|---|---|
| **76cc59efdcfe8789befb19e5544762e3** | Ascomycota | Dothideomycetes | Cladosporiaceae | *Cladosporium* | – |
| **ff75a588c4e4c84a13302714c9c099ef** | Ascomycota | Dothideomycetes | Cladosporiaceae | *Cladosporium* | – |
| **07ef2bd363cbb30aa9dd6502ce5031ab** | Ascomycota | Dothideomycetes | – | – | – |
| 163d9f9e5b068db5293a81186d8e8a36 | Ascomycota | Eurotiomycetes | Aspergillaceae | *Aspergillus* | *janus* |
| **dea354e5843dfbb801ab02cf52a47f6a** | Ascomycota | Eurotiomycetes | Aspergillaceae | *Aspergillus* | – |
| af8383f50d3fc377355ad0ae19b84da7 | Ascomycota | Saccharomycetes | Saccharomycetales | *Candida* | *parapsilosis* |
| 29bbc6dc5503c0663d8757c624d5116f | Ascomycota | Sordariomycetes | Plectosphaerellaceae | *Plectosphaerella* | *cucumerina* |
| <u>**36446a3379af55b9a5f64437440064c3**</u> | Ascomycota | Sordariomycetes | Nectriaceae | *Cylindrocladiella* | – |
| *78e8da310517f94e3b29daec7df59558* | Basidiomycota | Agaricomycetes | Hymenochaetaceae | *Hymenochaete* | – |
| *26a27a0b3a6843da2d387e6faa52529c* | Basidiomycota | Agaricomycetes | Meruliaceae | *Phlebiopsis* | *flavidoalba* |
| 5bfcfab69aa1f8e785a7f485dcac3f33 | Basidiomycota | Agaricomycetes | Steccherinaceae | *Nigroporus* | *vinosus* |
| **5489dcd887784454bf8c527dad7c95ec** | Basidiomycota | Malasseziomycetes | – | – | – |
| b9e7c6a8da7f304821968e590e203eda | Basidiomycota | Malasseziomycetes | Malasseziaceae | *Malassezia* | *restricta* |

**Table 2. Fungal ASVs from biofertilizer detected in forest and coffee soils.** If the fungal ASV was detected in any sample (e.g., drip line, upper) from a coffee bush then the fungal taxa was considered associated with that coffee bush. Two ASVs could be categorized into a trophic mode using FUNguild and our criterion for assignment: one was a 'probable' pathotroph (ASV ID underlined) and one was a 'probable' saprotroph (ASV ID italicized).

| ASV ID | Taxa | No. forest replicates (n = 10) | No. shade coffee bushes associated (n = 7) | No. sun coffee bushes associated (n = 7) |
|---|---|---|---|---|
| 76cc59efdcfe8789befb19e5544762e3 | *Cladosporium* sp. | 3 | 7 | 7 |
| ff75a588c4e4c84a13302714c9c099ef | *Cladosporium* sp. | 0 | 0 | 1 |
| 07ef2bd363cbb30aa9dd6502ce5031ab | Dothideomycetes sp. | 3 | 6 | 6 |
| dea354e5843dfbb801ab02cf52a47f6a | *Aspergillus* sp. | 0 | 1 | 1 |
| <u>36446a3379af55b9a5f64437440064c3</u> | *Cylindrocladiella* sp. | 10 | 3 | 0 |
| *78e8da310517f94e3b29daec7df59558* | *Hymenochaete* sp. | 0 | 0 | 1 |
| 5489dcd887784454bf8c527dad7c95ec | Malasseziomycetes sp. | 0 | 0 | 1 |

richness in the sun system when compared to the shade system (system effect: ANOVA, F (1,48) = 7.8, p = 0.007), which was primarily driven by the drip line samples (system x sampling location effect: ANOVA, F(1,48) = 5.9, p = 0.019). We found no effect of sampling depth on the richness of saprotrophs (ANOVA, p > 0.05).

## Sun vs. shade vs. forest dataset

We quantified the effect of system on alpha diversity and beta diversity for samples that were collected from the upper soil surface in sun and shade coffee systems and from a nearby forest.

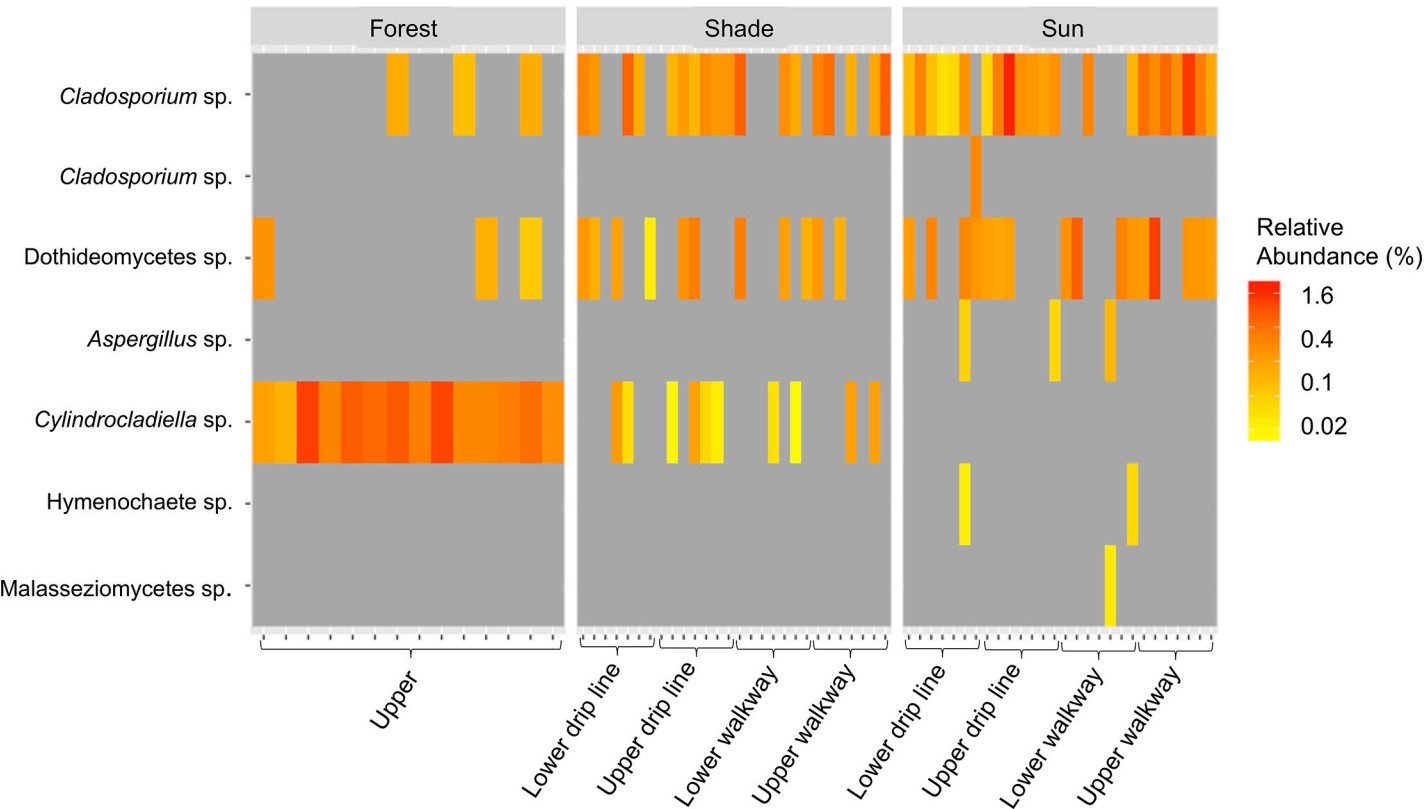

**Fig 3. Heatmap of biofertilizer fungal taxa found in forest and coffee soils.**

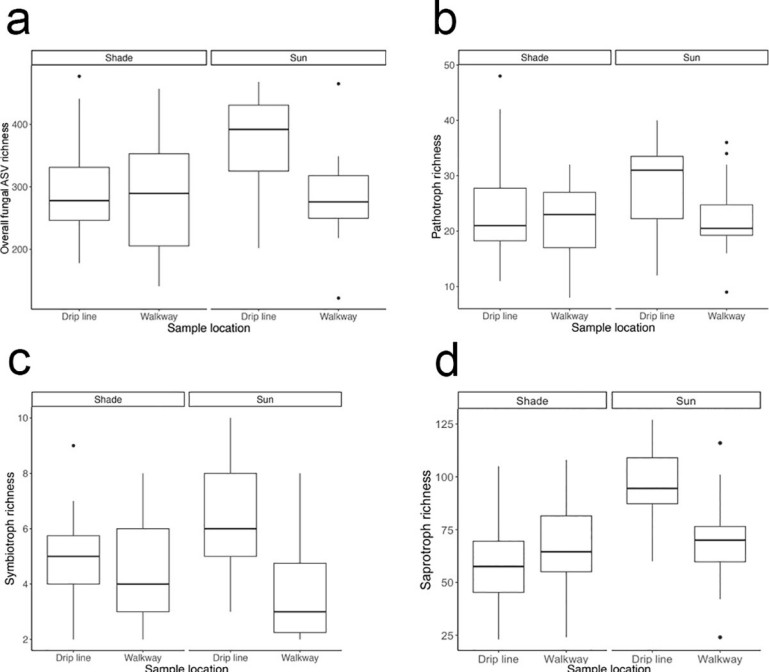

**Fig 4. Boxplots of overall fungal richness and trophic mode richness by system and sampling depth.** Each subfigure is plotted with system by sample location, but the interaction was only significant for subfigure d. Coffee soil collected in the drip line had higher a. overall fungal ASV richness, b. pathotroph richness, and c. symbiotroph richness than coffee soil collected in the walkway, regardless of system. d. Saprotroph richness was higher in the sun system when compared to the shade system, primarily driven by the drip line samples (interaction effect).

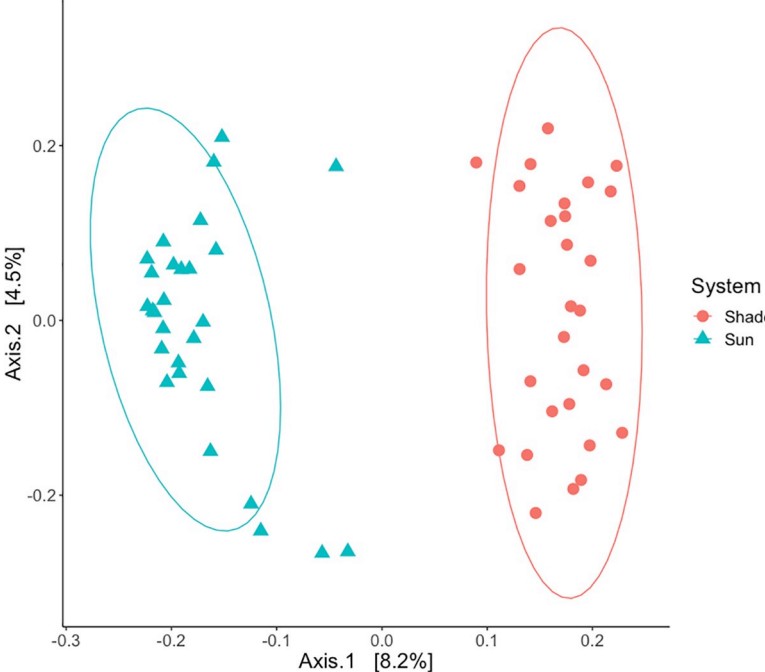

**Fig 5. PCoA visualizing Jaccard beta diversity patterns of sun and shade coffee systems.** Coffee soil collected from the sun-intensive system had distinct fungal communities from coffee soil collected in the traditional shade system.

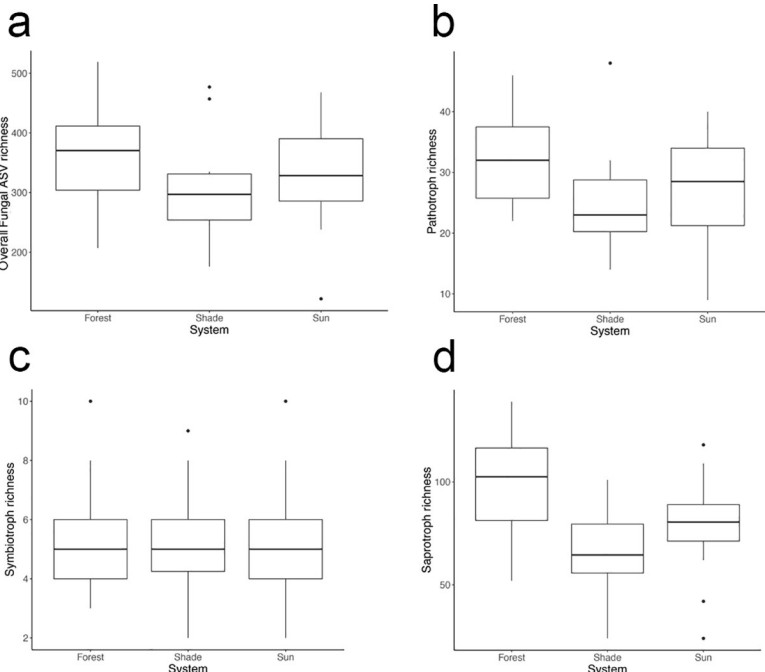

**Fig 6. Boxplots of overall fungal richness and trophic mode richness across sun, shade and forest systems.** Similar numbers of a. fungal taxa, b. pathotrophs and c. symbiotrophs were found in soil across systems. d. Saprotrophs richness was higher in the forest soil than in the traditional shade coffee system, and almost higher than the sun-intensive coffee system.

We found that ASV richness did not differ among systems (Fig 6A: ANOVA, F(2,39) = 2.2, p = 0.12). We found that Faith's phylogenetic diversity differed by system (S3 Fig: ANOVA, F(2,39) = 3.3, p = 0.048), with a trend of the forest system having higher phylogenetic diversity than the shade system (Tukey HSD: p adj = 0.069). Average soil fungal ASV richness was 335 ASVs (Fig 6A, SD ± 90) and average Faith's phylogenetic diversity was 72 tree units (S2 Fig, SD ± 16). Sun, shade and forest soil samples all showed distinct fungal community compositions (Fig 7: Jaccard: PERMANOVA, Pseudo F = 4.4, df = 2, R2 = 18.5%, p = 0.001; Bray-Curtis PERMANOVA, Pseudo F = 5.8, df = 2, R2 = 23.0%, p = 0.001). Pathotroph and symbiotroph richness was similar among sun, shade, and forest soils (Fig 6B and 6C: ANOVA, p > 0.05). Saprotroph richness differed among the three systems (Fig 6D: ANOVA, F(2,39) = 6.4, p = 0.004) with higher saprotroph richness in the forest than in the shade (Tukey HSD: p adj = 0.003).

We identified fungi that were unique and shared among the sun, shade, and forest systems, including 356 fungi found in all three (Fig 8). Sun and shade coffee soil had the most shared ASVs (641 ASVs, 22%, Fig 8). We used indicator species analysis to identify which of those fungi had the strongest specificity and fidelity to coffee soil indicating a likely symbiotic relationship with coffee plants. We found 20 fungal ASVs that were strongly associated with coffee plants (Table 3). These ASVs all belonged to the Ascomycota phyla and were predominantly in the classes Dothideomycetes and Sordariomycetes.

## Discussion

Shade coffee systems provide refuge for a larger variety of macro- and micro-organisms [49, 50]. Traditional shaded coffee systems often have higher species richness in macro-organisms

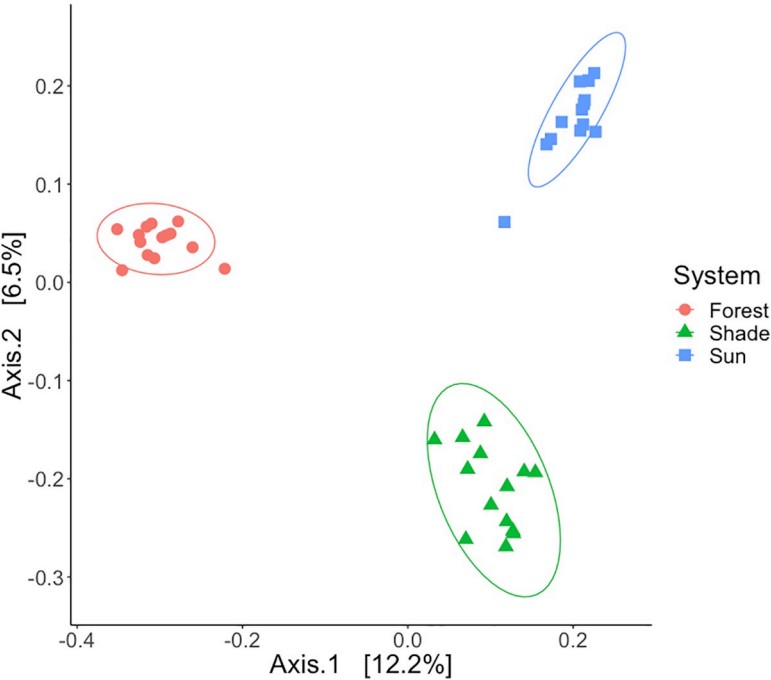

**Fig 7. PCoA visualizing Jaccard beta diversity patterns of sun, shade, and forest systems.** Fungal community composition in sun, shade, and forest systems were significantly different from one another.

such as pollinators [5], amphibians [11], and birds [9, 10] than sun-intensive coffee plantations. Yet, we did not detect any large differences in fungal richness between sun and shade coffee systems nor with the coffee systems compared to the forest soil. Macro-organisms may respond to different environmental cues than microorganisms. Cameron et al., [51] compared aboveground biodiversity with soil biodiversity including data on macrofauna, fungi, and bacteria from across the globe. Areas of mismatch between aboveground and soil biodiversity covered 27% of the world's terrestrial surface [51], and factors such as soil pH and climatic effects likely contributed to this deviation [52]. Biodiversity patterns of soil microorganisms may not follow the same diversity patterns as macrofauna.

Fungal community composition differed dramatically among sun coffee soil, shade coffee soil and forest soil, and this divergence may be attributed to differences in temperature, foliage and leaf litter and human impact. Because of their higher sun exposure, sun systems can be approximately 6°C higher than shade systems and forests [21]. Temperature and soil moisture often follow an inverse relationship, with soil moisture decreasing as temperature increases [53], and fungi tend to have optimal growth rates at around 25–30°C [54]. However, Bárcenas-Moreno et al., [55] found that over time, fungi could increase their temperature tolerance and shift their optimum temperatures above the standard 30°C. This response is likely due to a species sorting mechanism, where genetically advantaged fungal species adapted to higher temperatures outcompete less well-adapted species. This species sorting may influence the community composition, with different fungal taxa that are better adapted for higher temperatures living in sun systems, and other fungal taxa better adapted for lower temperatures living in the shade systems. Shade systems grow coffee under a large canopy of shade trees, which provides a layer of leaf litter [2]. Losses in aboveground biodiversity are known to directly impact litter decomposition through the loss of specific plant-soil interactions belowground [56]. Basidiomycota and Ascomycota, the two most predominant phyla of fungi found in our

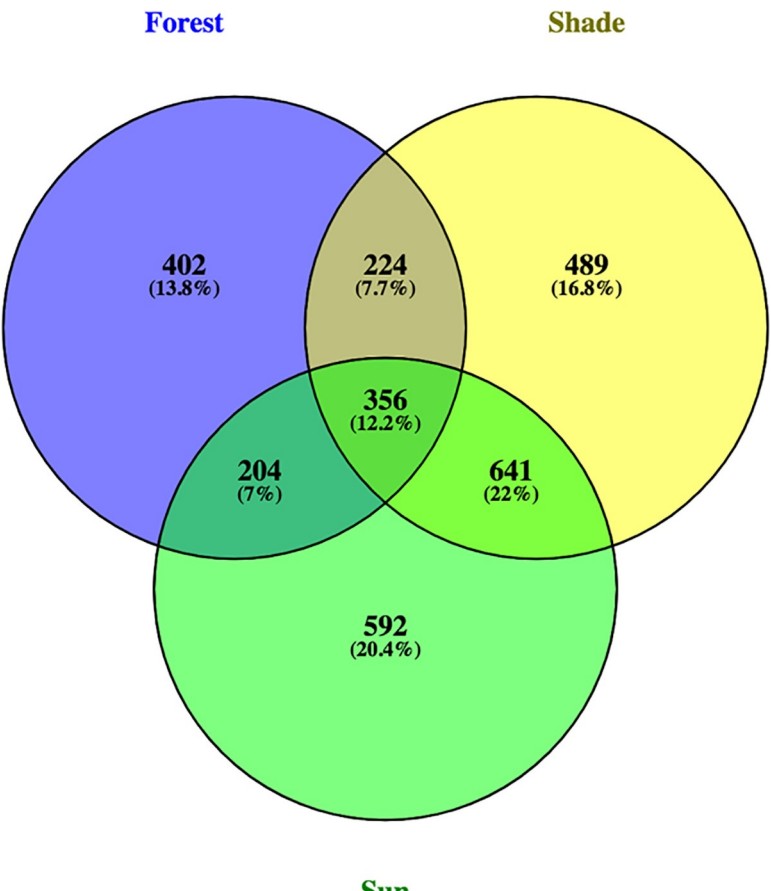

**Fig 8. Venn diagram of shared and unique fungal ASVs among the three systems.**

study, include genera like Mycena and Xylaria, which are frequently studied for their ability to decompose lignin in leaves [57]. In the forest, there is little human impact, including no fertilizer or other inputs, compared to coffee systems, which may explain the unique fungal community found in forest soils. Differences in temperature, leaf litter and human impact likely contribute to fungal species sorting among systems.

We found fungal taxa that have the potential to be screened for use in biofertilizers and for properties of plant-growth promoting fungi [58]. Specifically, we found (i) 356 fungal taxa that occurred in both coffee systems and in the nearby forest and (i) 7 fungal taxa that occurred in the currently used biofertilizer and in coffee soil. In the farms we sampled, farmers are already applying an organic biofertilizer derived from forest soil to increase yields in both sun and shade systems. We found that three of the thirteen fungal taxa in the biofertilizer were commonly detected in the coffee soil, suggesting that only certain fungi in the biofertilizer are effective at colonizing the soil associated with coffee plants. Alternatively, these fungi may already occur in the soil associated with coffee plants and their presence may not be a direct result of the biofertilizer being applied. Two of the dominant fungal taxa present in the biofertilizer, a *Cladosporium* sp. and a Dothideomycetes sp., showed greater relative abundance in the coffee soil when compared to the nearby forest where the biofertilizer is not applied, suggesting that the biofertilizer is increasing their abundance. Some *Cladosporium* sp. produce gibberellins, which play a vital role in plant growth and development, and can be considered plant-growth

**Table 3. Fungal ASVs strongly associated with the coffee bushes in both sun and shade systems and rarely found in the forest samples.** All fungi belong to the Ascomycota phylum. We used NCBI BLASTN 2.9.0+ to present greater taxonomic information, beyond what was assigned using QIIME2 (denoted by *), if greater taxonomic information was available. Four ASVs could be categorized into a trophic mode using FUNguild and our criterion for assignment, two were 'probable' pathotrophs (ASV ID underlined) and two were 'probable' saprotrophs (ASV ID italicized).

| ASV ID | Class | Order | Family | Genus | Species |
|---|---|---|---|---|---|
| 76cc59efdcfe8789befb19e5544762e3 | Dothideomycetes | Capnodiales | Cladosporiaceae | *Cladosporium* | – |
| ff9300e30824b54415be132c8c6b7945 | Dothideomycetes | Pleosporales | Cucurbitariaceae | *Pyrenochaetopsis* | *leptospora* |
| bf1ad896233b41b4951f910cbe244bc0 | Dothideomycetes | Pleosporales | Didymellaceae | *Ascochyta* | *medicaginicola* |
| a3128a218760aeffa35ac57bce74239d | Dothideomycetes | Pleosporales | Thyridariaceae* | – | – |
| eaef3faefb7628590db24e9b717349f5 | Dothideomycetes | Pleosporales* | Phaeosphaeriaceae | *Setophoma* | *terrestris* |
| de5fab791fe87470ac96b582b6f9bfb1 | Eurotiomycetes | Chaetothyriales | Herpotrichiellaceae | *Phialophora* | – |
| 9a780fddbe0f2142527dcbe6838d9541 | Eurotiomycetes | Eurotiales | Aspergillaceae | *Aspergillus** | *keveii* |
| be84a77a3b0890f2b2ad468f010b3141 | Eurotiomycetes* | – | – | – | – |
| 5c789cda30cf89599923ff5170bcee74 | Sordariomycetes | Glomerellales | Plectosphaerellaceae | *Plectosphaerella* | *cucumerina* |
| d972665f779bc48748e2beb0a9f98e4d | Sordariomycetes | Hypocreales | Nectriaceae | *Bisifusarium* | *dimerum* |
| 69cb98450ee65d6674647caf26727a27 | Sordariomycetes | Hypocreales | Nectriaceae | *Fusarium** | *oxysporum* |
| 2da753b0cbebd0db14b45d483b41d79c | Sordariomycetes | Hypocreales | Nectriaceae* | *Fusarium* | *buharicum* |
| bcc10feefbe0ce69b352d447ee068c67 | Sordariomycetes | Hypocreales | Nectriaceae* | *Fusarium* | *brachygibbosum* |
| debfecf0b2bbc97869b36022741bee50 | Sordariomycetes | Hypocreales | Nectriaceae* | *Fusicolla* | – |
| 14c6270556c788dfe5b20516e1966e16 | Sordariomycetes | Hypocreales | Stachybotryaceae* | *Albifimbria* | *verrucaria* |
| 7a911cf0806a4d8a89c1d667d2547d28 | Sordariomycetes | Microascales | Microascaceae* | *Lophotrichus* | *fimeti* |
| 566c1c04832f880a8fb506e797b724e2 | Sordariomycetes | Microascales* | Microascaceae | – | – |
| *3a10ab3e5b3d451189268a313e7b0230* | Sordariomycetes | Sordariales | Chaetomiaceae* | *Humicola* | *fuscoatra* |
| *9c62398b8b06f42a2a2a3403a96956a8* | Sordariomycetes | Sordariales | Sordariaceae* | *Sordaria* | *conoidea* |
| ab556a914eb0284b5c3beb81d0904c96 | Sordariomycetes | Sordariales* | – | – | – |

promoting fungi (PGPF) [59]. Future work to isolate cultures of candidate PGPF and conduct experiments on their functional capacity are warranted so that farmers looking to increase longevity of coffee in sun systems or increase coffee yields in shaded systems could inoculate soil at the coffee drip line with PGPF to more efficiently produce coffee.

We found twenty fungal taxa that we considered coffee plant-associated fungi (found in multiple samples in sun and shade coffee samples and almost exclusively in those samples). Coffee plants may influence properties of the soil and nutrient pool that selects for particular fungal taxa that are unique from those found in the forest habitat [60]. Microbial groups most directly associated with plant roots, like mycorrhizal fungi, have been found to exhibit a higher degree of specificity than previously supposed [61]. We also found evidence for more fungal taxa selecting for microhabitats in the drip line of the coffee as opposed to the walkway between two plants, supporting a fungal-coffee symbiotic relationship for many of the fungal taxa we detected. The majority of the coffee plant-associated fungal taxa did not have matches in sequence databases or could not be assigned to trophic mode. While some of the fungi detected as coffee-associated and also in the biofertilizer were placed into the trophic mode of pathotroph, these fungi are likely not active in this regard as no signs of disease were apparent on the plants. These findings underscore that the vast fungal diversity and their function in coffee soils is unknown.

Fungi from the ascomycete class Sordariomycetes were notably shared between the two coffee systems and included orders such as Hypocreales. One of the richest sources of biocontrol fungi have been from the order Hypocreales [62], with several strains commercially produced such as *Metarhizium* (Met52, Novozymes Biologicals) used as a bioinsecticide and *Trichoderma harzianum* (T-22 HC, BioWorks) used to control soil-borne diseases. We found a

*Cladosporium* sp. (ASV ID: 76cc59efdcfe8789befb19e5544762e3) to be present in the biofertilizer and strongly associated with coffee plants in both shaded and sun systems elevating this fungal taxa as a candidate for future study.

Ascomycota was the dominant phyla detected across all sample types. Either Ascomycota ASVs were more dominant in the mixed DNA sample or this result may be an artefact of amplification biases during PCR. However, previous research demonstrated that the primer pair we chose outperforms other primer pairs (targeting regions in the 18S and 28S genes and ITS1 and ITS2) in terms of efficiency and taxonomic coverage [32, 63], and that primers targeting the ITS1 and ITS2 detected roughly similar proportions of Ascomycota and Basidiomycota [64]. This provides support for Ascomycota being biologically dominant in the samples.

Surprisingly, we did not find a difference in fungal richness or community composition between soil depths (surface and 10 cm depth samples). The rhizosphere, the zone that surrounds the plant's roots, is dominated by fungi that are linked to plant health and growth [65]. AM fungal communities in deep soil layers are diverse and differ from those in the topsoil [66], and vertical stratification of soil should be taken into consideration when analyzing soil biodiversity [67]. However, the lack of variation in richness and community composition may be attributed to the coffee bush's root system. *Coffea arabica* roots are superficial, shallow, and extend more horizontally in comparison to other crop species [68]. Additionally, fine roots can be found in the litter layer of coffee plantations. The coffee plants in this study were relatively young at three to four years, so their roots did not extend very far vertically or horizontally. Future studies could look at coffee plantations with older plants and more established root structures to test for differences between surface and lower depth samples.

One of the most devastating issues crop farmers across the world face today is yield losses due to diseases caused by fungal pathogens. To combat pathogens in other crops and plant species, researchers have successfully identified antagonistic microbes as biological controls to target the disease-causing pathogens [69]. For instance, Xue et al., [30] found that the application of the bacteria *Bacillus* (dominant in disease-suppressive soil) altered the rhizo-bacterial community and helped to decrease pathogen colonization by Panama disease (*Fusarium oxysporum* f. sp. *cubense*) in the banana rhizosphere. Coffee leaf rust and coffee wilt disease (tracheomycosis) are the two primary fungal pathogens affecting coffee plants across Africa and Central and South America. While there is currently no known cure for these diseases, researchers tested various strains of rhizobacteria isolates and found the first evidence of coffee-associated rhizobacteria having antagonistic effects on fungal coffee pathogens [70]. Using symbiotic fungi to promote plant growth and fight infection has the potential to shift current coffee practices from using harmful chemical pesticides to more ecologically friendly microbial inoculates. Our study offers insight into the diversity and distribution of fungi in coffee soil laying the groundwork for more directed studies on fungi (such as *Cladosporium* sp. [ASV ID: 76cc59efdcfe8789befb19e5544762e3]) that can contribute to plant growth promotion and disease resistance.

Over the past 20 years, researchers at the Smithsonian Migratory Bird Center (SMBC) have assessed the biological impact of shade cover in coffee versus sun-intensive plantations. These efforts have highlighted that traditional shade coffee systems provide viable habitat for taxa like birds, insects and mammals, including the early studies by Greenberg, which was the basis for the SMBC's creation of a shade certification for coffee—the Bird Friendly program (si.edu/smbc). This research has also focused on the socio-economic benefits of coffee agroforestry systems—in particular the non-coffee products like fruits and wood that farmers can use or sell [71–73]. More recently, the ability of a shaded system to combat a number of the challenges that coffee farmers face due to climate disruption/change has also been discussed [72]. A multi-country effort to determine which shade trees used by producers yield the most food

resources in the form of fruits and/or insects has begun to identify those trees species that could help enhance the ecological health of coffee agroforestry systems [74]. The findings of the present study reveal that fungal communities also differ between sun and shade coffee plantations, with many symbiotrophs living in coffee soil, which have the potential to be used as probiotic inoculants to increase crop yields in the agroforestry system. Together, these studies—whether focusing on the soil mycobiome or the habitat dimensions of the shade coffee site—add to our general knowledge of how these agroforestry systems can potentially aid coffee farmers with interest in enhancing the sustainable aspects of their holdings.

## Supporting information

**S1 Fig. The 'mountain fungus' is collected from leaf litter in forests nearby to coffee plantations and is used to make the biofertilizer.**
(DOCX)

**S2 Fig. Boxplots of overall fungal phylogenetic diversity by system and sampling depth in sun and shade coffee soil.**
(DOCX)

**S3 Fig. Boxplots of overall fungal phylogenetic diversity across sun, shade and forest systems.**
(DOCX)

## Acknowledgments

We would like to thank Dr. Alexandra Bely and Dr. Nathan Swenson for their advice and support while working on this project. We also thank Robert and Arlene Kogod for their contribution to the Smithsonian Secretarial Scholars program (supporting CRM-W), as well as the staff of CLUSA El Salvador and the members of the ACOPRA Las Lajas Coffee Cooperative.

## Author Contributions

**Conceptualization:** Robert A. Rice, Robert C. Fleischer, Carly R. Muletz-Wolz.

**Formal analysis:** Maya V. Rao, Carly R. Muletz-Wolz.

**Investigation:** Maya V. Rao.

**Methodology:** Maya V. Rao, Robert A. Rice, Carly R. Muletz-Wolz.

**Supervision:** Robert C. Fleischer, Carly R. Muletz-Wolz.

**Visualization:** Maya V. Rao, Carly R. Muletz-Wolz.

**Writing – original draft:** Maya V. Rao, Carly R. Muletz-Wolz.

**Writing – review & editing:** Maya V. Rao, Robert A. Rice, Robert C. Fleischer, Carly R. Muletz-Wolz.

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
