## [Decision Letter · Decision Letter 0]

31 Oct 2019

PONE-D-19-22458

Soil fungal communities differ between shaded and sun-intensive coffee plantations in El Salvador

PLOS ONE

Dear Dr. Muletz-Wolz,

Thank you for submitting your manuscript to PLOS ONE. Please, first let me apologize for the delay in handling your manuscript. This was due to the fact that two referees successively accepted to review the paper but did not return their review in due time... I thus decided to base my decision on the single review that I could obtain as well as the opinion of an expert biostatistician for the data analysis part. After careful consideration, we feel that it has merit but does not fully meet PLOS ONE’s publication criteria as it currently stands. Therefore, we invite you to submit a revised version of the manuscript that addresses the points raised during the review process.

 As you will see in the referee’s review, he/she has some questions regarding the details provided in the manuscript. I do concur with the referee that more info should be given in the M & M on how you performed the analyses, but also in the figure legend. Especially, statistical supports of the differences between the samples could be indicated in the figure as they are tedious to read in the text. I would also urge you to analyze in greater details the species that you identified. This is especially important for the analysis of the Sordariomycetes as they represent nearly half the ASVs in all your samples. You could classify them in the different orders, as some contain a lot of plant parasites (e.g., Hypocreales), while other do not (e.g. Sordariales).  Providing the closest species in table 1 for ASVs  presently missing one should also be a bonus. The biostatistician has the following additional comments:

Methods:  Did the authors use default parameters of FastTree for phylogenetic tree analysis (e.g., JC69 model)?

Results:

- Fig 2 It would be more helpful to keep figure legend as informative as possible regarding their content. The second part of this legend should probably go in the body of the text. The same remark applies to Fig 5, 6 and 8. Note that Figure 2 also summarizes results from sampling depth but this is not discussed in the text.

- [Sun vs. shade dataset] If average value for ASV richness is reported, then average value for Faith's PD may also be summarized. Figures 3a and 3b show almost the same pattern for the qualitative interaction, so the trend for an interaction in the case of ASV richness as opposed to Faith's PD is probably spurious, as confirmed by the associated p-values which are largely above 0.05. I would also suggest to limit the number of decimal places to 1 for R2 values, and to use a consistent precision for F-values (e.g., one decimal for one-way conventional ANOVA but 5 significant digits for permutational ANOVA). Finally, sampling depth is only discussed with regard to community composition, as far as I can tell. No results on ASV richness and phylogenetic diversity are given. Why is it so? The same remark applies to the comparison of the 3 clusters (pathotrophs, symbiotrophs, and saprotrophs). It is quite surprising since the absence of difference in terms of fungal richness is mentioned upfront in the abstract and in the conclusion (l.383 ff.).

- [Sun vs. shade vs. forest dataset] Shouldn't the numerator DF of the one-way F-tests (l.274 and 276) be 2 since there are 3 conditions? Post-hoc Tukey analysis should be mentioned in the Methods section.

- Fig 6a Isn't the upper whisker for the shade condition somewhat missing, or is this just a printing issue with the PDF reprint?

He/she also detected the following Typos:

- l.105: elevation of

- l.128: he should probably read we

- l.224: "effect of system, sampling depth, sampling location" it would be better to put this in the same order as the statistical analyses discussed below, i.e. "effect of system, sampling location, sampling depth"

- l.250: sampling locations should read sampling location

- l.283: with saprotroph richness higher should read with higher saprotroph richness

- l.300-302: please rephrase this sentence

- l.322: except saprotroph should read except that saprotroph

We would appreciate receiving your revised manuscript by Dec 15 2019 11:59PM. To enhance the reproducibility of your results, we recommend that if applicable you deposit your laboratory protocols in protocols.io, where a protocol can be assigned its own identifier (DOI) such that it can be cited independently in the future. For instructions see: http://journals.plos.org/plosone/s/submission-guidelines#loc-laboratory-protocols

We look forward to receiving your revised manuscript.

Kind regards,

Philippe Silar

Academic Editor

PLOS ONE

Journal Requirements:

Reviewers' comments:

Reviewer's Responses to Questions

**Comments to the Author**

1. Is the manuscript technically sound, and do the data support the conclusions?

Reviewer #1: Yes

2. Has the statistical analysis been performed appropriately and rigorously? 

Reviewer #1: Yes

3. Have the authors made all data underlying the findings in their manuscript fully available?

Reviewer #1: Yes

4. Is the manuscript presented in an intelligible fashion and written in standard English?

Reviewer #1: Yes

5. Review Comments to the Author

Reviewer #1: The authors did a great assessment of the diversity in soil with coffee plantation under shaded and sun intensive area in El Salvador. They use metagenomics with ITS2 region to monitor the diversity. They used QIIME2 and other software to evaluated it. I think they did great job to evaluate diversity but still missing some details, references in M&M and a bit more search on species level before to made those conclusion. They seems to have done a great bioinformatics work but still need to do some for improving the manuscript. Did you really have answer all your 3 aims of the introduction with the functional guild? P5. How the FUNGuild was used reference?

M&M field sampling any replicates used?

P7L143 any references about the primers barcoded i5 i7?

You mentioned kits but no info on kit company ? exP8 SPRI-beads?

P9 can add more info on what is the diversity of Faith'sPD?

You use ITS sequences, did you used ITSX to extract only those sequences? It was not clearP10 and M&M what are the reference database you are using. Usually we are use UNITE for Fungi. How the determination of some species was done not clear? what Phylum is usually get because not many in Table 1? Also not sure it was well explained how the species or ASB related to Pathotroph, symbiotrop and saprotroph were assigned and identified, not clear? Sometine ITS2 and ITS 1 not similar value, maybe couls explain some of the difference un asco and basidio? The figures titles can be rework " ex significantly impacted, should be in discussion not in figure title?Fig 3-4-7? Does this manuscript present all the potential of analysis of this work with only presenting richness and diversity value, could present more, please look at other publications? Fig 9 done at which phylum level? Table 1 only few ID? why?

6. PLOS authors have the option to publish the peer review history of their article (what does this mean?). If published, this will include your full peer review and any attached files.

Reviewer #1: No

---

## [Author Response · Author response to Decision Letter 0]

31 Mar 2020

Response to reviewer: 

Editor’s response: Thank you for submitting your manuscript to PLOS ONE. Please, first let me apologize for the delay in handling your manuscript. This was due to the fact that two referees successively accepted to review the paper but did not return their review in due time... I thus decided to base my decision on the single review that I could obtain as well as the opinion of an expert biostatistician for the data analysis part. After careful consideration, we feel that it has merit but does not fully meet PLOS ONE’s publication criteria as it currently stands. Therefore, we invite you to submit a revised version of the manuscript that addresses the points raised during the review process.

As you will see in the referee’s review, he/she has some questions regarding the details provided in the manuscript. I do concur with the referee that more info should be given in the M & M on how you performed the analyses, but also in the figure legend. Especially, statistical supports of the differences between the samples could be indicated in the figure as they are tedious to read in the text. I would also urge you to analyze in greater details the species that you identified. This is especially important for the analysis of the Sordariomycetes as they represent nearly half the ASVs in all your samples. You could classify them in the different orders, as some contain a lot of plant parasites (e.g., Hypocreales), while other do not (e.g. Sordariales). Providing the closest species in table 1 for ASVs presently missing one should also be a bonus. 

Our response: Thank you for your time in looking at our manuscript. We have updated the figures and figure legends to provide more detail. We added more taxonomic resolution to Table 1 using ncbi blastn searches. Given your suggestion, we have analyzed in greater details the species we identified, particularly Sordariomycetes (Lines 441-449) and also added information about the biofertilizer used in the system. We added to the text the average relative abundance of Sordariomycetes and Dothideomyctes. 

The biostatistician has the following additional comments:

Methods: Did the authors use default parameters of FastTree for phylogenetic tree analysis (e.g., JC69 model)?

Our response: We did use the default parameters for FastTree. 

Results:

- Fig 2 It would be more helpful to keep figure legend as informative as possible regarding their content. The second part of this legend should probably go in the body of the text. The same remark applies to Fig 5, 6 and 8. Note that Figure 2 also summarizes results from sampling depth but this is not discussed in the text.

Our response: Thank you for your suggestion. We have updated the figures and all of our figure legends to be more informative. 

- [Sun vs. shade dataset] If average value for ASV richness is reported, then average value for Faith's PD may also be summarized. Figures 3a and 3b show almost the same pattern for the qualitative interaction, so the trend for an interaction in the case of ASV richness as opposed to Faith's PD is probably spurious, as confirmed by the associated p-values which are largely above 0.05. 

Our response: We included a summary of Faith’s PD average values now. We removed the text about the trend for an interaction.

I would also suggest to limit the number of decimal places to 1 for R2 values, and to use a consistent precision for F-values (e.g., one decimal for one-way conventional ANOVA but 5 significant digits for permutational ANOVA). 

Our response: We consistently now use one decimal place for all F-values. 

Finally, sampling depth is only discussed with regard to community composition, as far as I can tell. No results on ASV richness and phylogenetic diversity are given. Why is it so? The same remark applies to the comparison of the 3 clusters (pathotrophs, symbiotrophs, and saprotrophs). It is quite surprising since the absence of difference in terms of fungal richness is mentioned upfront in the abstract and in the conclusion (l.383 ff.).

Our response: Thank you for pointing that out. We only reported results when there was a significant or near significant effect. We have since updated the results to be more explicit on when what factors and interactions were significant or not. This applies to sampling depth that is now discussed in the results.

- [Sun vs. shade vs. forest dataset] Shouldn't the numerator DF of the one-way F-tests (l.274 and 276) be 2 since there are 3 conditions? Post-hoc Tukey analysis should be mentioned in the Methods section.

Our response: Sorry, that was a mistake on our part. Yes, the numerator DF was 2, and has been updated. We have added to the methods – ‘We conducted post hoc analyses for significant terms using the Tukey’s ‘Honest Significant Difference’ (HSD) method.’

- Fig 6a Isn't the upper whisker for the shade condition somewhat missing, or is this just a printing issue with the PDF reprint? 

Our response: No, that is the true upper whisker length. It does look short, but we verified that it is not a visual issue. It likely is a reflection of the two outlier points then shortening the upper whisker – the next third largest data point is not far in value from the 3rd quartile. 

He/she also detected the following Typos:

- l.105: elevation of

Our response: Corrected.

- l.128: he should probably read we

Our response: Corrected.

- l.224: "effect of system, sampling depth, sampling location" it would be better to put this in the same order as the statistical analyses discussed below, i.e. "effect of system, sampling location, sampling depth"

Our response: We now consistently use effect of system, sampling depth, sampling location throughout the entire manuscript.

- l.250: sampling locations should read sampling location

Our response: Corrected.

- l.283: with saprotroph richness higher should read with higher saprotroph richness

Our response: Corrected.

- l.300-302: please rephrase this sentence

Our response: We apologize for missing the typos in that sentence. It now reads ‘We identified fungi that were unique and shared among the sun, shade, and forest systems, including 356 fungi found in all three (Fig 8).’

- l.322: except saprotroph should read except that saprotroph

Our response: Corrected.

Reviewer #1: The authors did a great assessment of the diversity in soil with coffee plantation under shaded and sun intensive area in El Salvador. They use metagenomics with ITS2 region to monitor the diversity. They used QIIME2 and other software to evaluated it. I think they did great job to evaluate diversity but still missing some details, references in M&M and a bit more search on species level before to made those conclusion. They seems to have done a great bioinformatics work but still need to do some for improving the manuscript. 

Our response: Thank you for a favorable review of our research. We are glad that you found the paper to be of great quality. We appreciate your suggestions to improve the manuscript. We have added more details in the M & M and more information of the species we identified in the study. We have also sequenced the biofertilizer that was applied to the coffee soil systems to add more to our conclusions about the system. 

Did you really have answer all your 3 aims of the introduction with the functional guild? P5. 

Our response: We chose to do analyses on the entire community and not just those assigned to functional guilds for the first two aims. The main reason for this is that we do not know the functional guilds of approximately 50% of the fungi we discovered in the soil samples. That is their taxonomic assignments were not sufficient to be assigned a guild in FUNGuild, or they were eliminated based on our criterion we specify in the M&M on lines 192-197. 

How the FUNGuild was used reference? 

Our response: We followed the exact protocol of FUNguild and followed citation [41] on only including fungi that had hits as probable or highly probable. We have updated the text to include more information about the procedure we followed on line 192, which reads:

‘We uploaded an ‘OTU file’, which included taxonomic assignments for each ASV to the online classifier for FUNguild. ASVs were assigned to trophic modes (pathotrophs, symbiotrophs, and saprotrophs) if there was sufficient taxonomic information for the ASV.’

M&M field sampling any replicates used? 

Our response: Yes, this was in the original text on line 110. To be more clear on sampling we added an additional sentence on line 126, which reads: 

‘In total, we had one sample of the biofertilizer and 70 soil samples (Fig 1; 28 samples from seven sun coffee bushes, 28 samples from seven shade coffee bushes, 14 samples from a nearby forest).’

P7L143 any references about the primers barcoded i5 i7?

Our response: We have clarified that they were nextera-style i5 and i7 adaptors in the text. We ordered nextera adaptors from ‘Illumina Adapter Sequences’ and dilute our own to use for multiple projects as opposed to buying a nextera index kit, which is much more expensive. 

You mentioned kits but no info on kit company ? exP8 SPRI-beads?

Our response: Thank you for noticing that. We make our own beads in house that are SPRI-bead substitutes. We have included the proper term – speed beads – for them and added the citation on making them, which is available in the supplement of the paper. 

‘We performed post-PCR clean-ups between each PCR reaction using Speed-beads (in a PEG/NaCl buffer). ‘

P9 can add more info on what is the diversity of Faith'sPD?

Our response: We had added this to the M&M on line 187:

‘Faith’s PD is the phylogenetic analogue of taxon richness and is expressed as the number of tree units which are found in a sample.’

You use ITS sequences, did you used ITSX to extract only those sequences? It was not clear

Our response: No, we did not use ITSX. Thank you for bringing that program to our attention. Instead we examined the taxonomic classifications and removed any ASVs that were not assigned as fungi. We have now included that in the text: 

‘We removed all ASVs that were not assigned to fungi at the kingdom level.’

P10 and M&M what are the reference database you are using. Usually we are use UNITE for Fungi. How the determination of some species was done not clear? what Phylum is usually get because not many in Table 1? 

Our response: We used UNITE as indicated on line 169. We used QIIME 2 and the following code, which in some cases could assign species level classifications to ASV sequences. 

qiime feature-classifier classify-sklearn \\

 --i-classifier unite-ver7-99-classifier-01.12.2017.qza \\

 --i-reads rep-seqs-coffee1.qza \\

 --o-classification taxonomy-coffee.qza

We included in Table 1 legend now: ‘All fungi belong to the Ascomycota phylum.’

We added more taxonomic resolution to Table 1 using ncbi blastn searches.

Also not sure it was well explained how the species or ASB related to Pathotroph, symbiotrop and saprotroph were assigned and identified, not clear? 

Our response: Thank you. Using FUNGuild, we assigned pathotroph, symbiotroph, and saprotroph to the ASVs. As indicated above we included more to the M&M on the procedure lines 186-191, which now read:

‘We used FUNGuild [39] to assign fungal ASVs to specific functions, which may include aiding in nutrient uptake and protecting against pathogens [40]. We uploaded an ‘OTU file’, which included taxonomic assignments for each ASV to the online classifier for FUNguild. ASVs were assigned to trophic modes (pathotrophs, symbiotrophs, and saprotrophs) if there was sufficient taxonomic information for the ASV. We included only “probable” and “highly probable” matches (discarding “possible” matches), and filtered for ASVs that exclusively were assigned as symbiotrophs, pathotrophs, and saprotrophs [41].’

Sometine ITS2 and ITS 1 not similar value, maybe couls explain some of the difference un asco and basidio?

Our response: We used primers (ITS86F and ITS4) targeting ITS2. We did a thorough literature review prior to choosing primers and chose those primers and the ITS2 for various reasons. Mainly, the primer pair we chose has been shown to outperform other primer pairs (targeting regions in the 18S and 28S genes and ITS1 and ITS2) in terms of efficiency and taxonomic coverage (Waud et al. 2014, Op De Beeck et al. 2014). Additionally, Blaalid et al. 2012 show that primers targeting the ITS1 and ITS2 detected roughly similar proportions of Ascomycota and Basidiomycota. Either Ascomycota ASVs are more dominant in the mixed DNA sample or this result may be an artefact of amplification biases during the PCR. Findings from Blaalid et al. 2012 provide support for Ascomycota being biologically more dominant. 

We have now added text on lines 455-462 re-iterating this point as above. 

The figures titles can be rework " ex significantly impacted, should be in discussion not in figure title?Fig 3-4-7?

Our response: We have updated all of the figure legends to first present what the reader is looking at as the title (e.g., boxplot of trophic mode richness by system), and then what the results are for that figure (e.g., saprotroph richness did not differ by system).

Does this manuscript present all the potential of analysis of this work with only presenting richness and diversity value, could present more, please look at other publications?

Our response: Thank you. We have looked at other publications. In addition to richness and diversity values, we present information on core fungal taxa found in coffee soil (Table 1). We have added more discussion about the fungi we detected such as on lines 411-426 and 440-454. This includes more information about the biofertilizer used, which is collected from forest soils. 

Fig 9 done at which phylum level? 

Our response: Fig 9 (now Fig 8) presents data on ASVs and not phyla. Fig 9 is showing the number of shared and unique fungal ASVs.

Table 1 only few ID? why? 

Our response: These were 20 fungal ASVs that were found to be strongly associated with both sun and shade coffee plants, and rarely found in forest soil. They were identified using indicator species analysis. The idea was to identify fungi that are likely forming an association with the coffee plants specifically, and not just general members of soil communities. We have since used NCBI blastn to increase the taxonomic resolution for some of the taxa in Table 1. They are denoted with an * now.

---

## [Editor Report · Decision Letter 1]

3 Apr 2020

Soil fungal communities differ between shaded and sun-intensive coffee plantations in El Salvador

PONE-D-19-22458R1

Dear Dr. Muletz-Wolz,

We are pleased to inform you that your manuscript has been judged scientifically suitable for publication and will be formally accepted for publication once it complies with all outstanding technical requirements.

With kind regards,

Philippe Silar

Academic Editor

PLOS ONE
---

## [Editor Report · Acceptance letter]

9 Apr 2020

PONE-D-19-22458R1 

Soil fungal communities differ between shaded and sun-intensive coffee plantations in El Salvador 

Dear Dr. Muletz-Wolz:

I am pleased to inform you that your manuscript has been deemed suitable for publication in PLOS ONE. Congratulations! Your manuscript is now with our production department. 

With kind regards,

on behalf of

Dr. Philippe Silar 

Academic Editor

PLOS ONE